# Awareness, Preference, and Acceptance of HPV Vaccine and Related Influencing Factors Among Guardians of Adolescent Girls in China: A Health Belief Model-Based Cross-Sectional Study

**DOI:** 10.3390/vaccines13080840

**Published:** 2025-08-06

**Authors:** Shuhan Zheng, Xuan Deng, Li Li, Feng Luo, Hanqing He, Ying Wang, Xiaoping Xu, Shenyu Wang, Yingping Chen

**Affiliations:** 1School of Public Health, Hangzhou Medical College, Hangzhou 310053, China; 130232023412@hmc.edu.cn; 2Department of Immunization Program, Zhejiang Provincial Center for Disease Control and Prevention, Hangzhou 310051, China; xdeng@cdc.zj.cn (X.D.); hanqinghe@cdc.zj.cn (H.H.); ywang@cdc.zj.cn (Y.W.); xpxu@cdc.zj.cn (X.X.); 3Department of Immunization Program, Chinese Center for Disease Control and Prevention, Beijing 102206, China; lili@chinacdc.cn; 4School of Public Health, Zhejiang Chinese Medical University, Hangzhou 310053, China; fengluo12310436@163.com; 5Zhejiang Key Lab of Vaccine, Infectious Disease Prevention and Control, Zhejiang Provincial Center for Disease Control and Prevention, Hangzhou 310051, China

**Keywords:** HPV vaccine, acceptance, adolescent girls, guardians, HBM

## Abstract

Background: Cervical cancer poses a threat to the health of women globally. Adolescent girls are the primary target population for HPV vaccination, and guardians’ attitude towards the HPV vaccine plays a significant role in determining the vaccination status among adolescent girls. Objectives: This study aimed to explore the factors influencing guardians’ HPV vaccine acceptance for their girls and provide clues for the development of health intervention strategies. Methods: Combining the health belief model as a theoretical framework, a questionnaire-based survey was conducted. A total of 2157 adolescent girls and their guardians were recruited. The multivariable logistic model was applied to explore associated factors. Results: The guardians had a high HPV vaccine acceptance rate (86.7%) for their girls, and they demonstrated a relatively good level of awareness regarding HPV and HPV vaccines. Factors influencing guardians’ HPV vaccine acceptance for girls included guardians’ education background (OR = 0.57, 95%CI = 0.37–0.87), family income (OR = 1.94, 95%CI = 1.14–3.32), risk of HPV infection (OR = 3.15, 95%CI = 1.40–7.10) or importance of the HPV vaccine for their girls (OR = 6.70, 95%CI = 1.61–27.83), vaccination status surrounding them (OR = 2.03, 95%CI = 1.41–2.92), awareness of negative information about HPV vaccines (OR = 0.59, 95%CI = 0.43–0.82), and recommendations from medical staff (OR = 2.32, 95%CI = 1.65–3.25). Also, guardians preferred to get digital information on vaccines via government or CDC platforms, WeChat platforms, and medical knowledge platforms. Conclusions: Though HPV vaccine willingness was high among Chinese guardians, they preferred to vaccinate their daughters at the age of 17–18 years, later than WHO’s recommended optimal age period (9–14 years old), coupled with safety concerns. Future work should be conducted based on these findings to explore digital intervention effects on girls’ vaccination compliance.

## 1. Introduction

Cervical cancer, the fourth most prevalent cancer among women, has emerged as a significant global public health concern [1]. According to Global Cancer Statistics 2022, approximately 660,000 new cervical cancer cases and 350,000 associated deaths occurred worldwide [2]. In China, both the incidence and mortality of cervical cancer have been increasing in recent years, accompanied by a trend toward a younger age of onset [3]. In 2022, China recorded an estimated 151,000 new cases (incidence rate: 13.8/100,000) and 56,000 deaths (mortality rate: 4.5/100,000), imposing a substantial burden on women’s health and social economy [4,5].

Persistent infection with high-risk human papillomavirus (HPV) is the primary etiological factor in cervical cancer, with 71% of cervical cancer cases attributable to persistent infection with HPV 16 and 18 [6]. HPV vaccination represents the most critical intervention for preventing cervical cancer and HPV-associated diseases [7]. WHO recommends prioritizing HPV vaccination for adolescent girls prior to sexual exposure and setting a target of achieving a 90% full HPV vaccination rate among girls under 15 by 2030 [8]. As of April 2025, 147 countries have included the HPV vaccine into their National Immunization Programs (NIPs) [9].

At present, six HPV vaccines have been approved for use in China: bivalent Cervarix (approved in 2016, 81 USD per dose), quadrivalent Gardasil (approved in 2017, 111 USD per dose), nine-valent Gardasil 9 (approved in 2018, 181 USD per dose), bivalent Cecolin (approved in 2019, 46 USD per dose), bivalent Walrinvax (approved in 2022, 45 USD per dose), and nine-valent Cecolin 9 (approved in 2025, 70 USD per dose). The HPV vaccine has not yet been included in the Chinese NIP, and the vaccination rate among adolescent girls remains at a relatively low level [10]. As adolescent girls are typically financially dependent, the majority are unable to afford HPV vaccination independently. Consequently, guardians’ awareness of HPV and their attitudes towards the HPV vaccine are pivotal in shaping adolescent girls’ vaccine acceptance [11]. A comparative analysis of two meta-analyses of parents’ HPV vaccine acceptance for adolescents in mainland China over different periods indicates a notable increase in HPV awareness, HPV vaccine awareness, and HPV vaccine acceptance rates, which now stand at 45.0%, 41.4%, and 61.0%, respectively [12]. Nevertheless, compared to some developed nations, the understanding of HPV and attitudes towards the HPV vaccine among parents of adolescents in mainland China remain suboptimal [13].

Previous research has demonstrated that the guardians’ HPV vaccine acceptance for girls is influenced by multiple determinants, with the guardians’ knowledge, practices, and preferences being key contributing factors [14]. The Health Belief Model offers a robust framework for predicting guardians’ beliefs and understanding of HPV and HPV vaccines, elucidating the underlying reasons for changes or maintenance of their attitudes towards HPV vaccination [15]. The Theory of Planned Behavior effectively describes guardians’ attitudes towards the HPV vaccine, subjective norms, and perceived behavioral control, serving as a guiding principle for exploring their intentions and behaviors [16]. Therefore, developing a Health Belief Model combined with the Theory of Planned Behavior could elucidate the factors influencing guardians’ HPV vaccine acceptance precisely.

This study aimed to (1) assess guardians’ awareness of HPV and HPV vaccines; (2) evaluate acceptance of HPV vaccine and related influencing factors among guardians of 9–17 adolescent girls; (3) provide evidence for the development of more tailored and evidence-based health interventions to enhance the actual HPV vaccination coverage among adolescent girls.

## 2. Materials and Methods

### 2.1. Study and Participants

We conducted a cross-sectional study from May to September 2024 in Zhejiang province, eastern China; participants were recruited through a multi-stage stratified cluster random sampling method. In the first stage, we selected three cities representing distinct socioeconomic levels in Zhejiang Province (Hangzhou, Jinhua, and Quzhou). In the second stage, four districts or counties were selected from each city, where the population, economic development level, and the baseline HPV vaccination rate among girls aged 9 to 17 were comparable. In the third stage, four vaccination clinics were selected from each district or county. Participants were recruited through vaccination clinics, and inclusion criteria were as follows: (1) guardians had at least one girl aged 9 to 17 and resided in their district or county for at least 6 months; (2) the girls had not received the HPV vaccine previously.

The sample size of this study was calculated by the following formula:
$$n=deff\times {\left(\frac{Z_{1-\alpha /2}}{\delta }\right)}^{2}\times p\times (1-p)$$

According to previous literature [11], we set the expected acceptance of HPV vaccine at P = 50%, with α = 0.05, δ = 0.04, deff = 2.5; the minimum required sample size was calculated as 1500. Considering a potential 10% non-response rate, the total target sample size was adjusted to 1650.

This study has been reviewed and approved by the Ethics Committee of the Chinese Center for Disease Control and Prevention (No. 202407), and all respondents provided informed consent and voluntarily filled out the questionnaires.

### 2.2. Procedures

The questionnaire passed five rounds of expert evaluation (including CDC experts, community health service center staff, and guardian representatives). All investigators received standardized training. Prior to study implementation, a pilot study with 32 participants was conducted in Hangzhou. To enhance participants’ comprehension of the questionnaire, revisions were made based on feedback from pilot participants (including shortening the length and rephrasing technical questions into more accessible language). These 32 participants were not involved in the actual survey. During the survey process, quality control staff provided on-site supervision. After investigators provided a standardized explanation, the questionnaires were completed on-site to guarantee the authenticity and validity of the collected questionnaire information.

### 2.3. Measures

#### 2.3.1. Health Belief Model

In our present study, the Health Belief Model and Theory of Planned Behavior were used as the theoretical framework to assess guardians’ HPV vaccine acceptance for their girls. The Health Belief Model in our study comprised three dimensions: Guardians’ HPV Perceptions (3 items), Guardians’ Vaccine Attitude (4 items), and Cues to Action (4 items). The Health Belief Model is shown in Figure 1.

#### 2.3.2. Data Collection

In our present study, the questionnaire collected data on (1) Demographic Factors, including children’s characteristics, such as age, education level, and household registration province, and guardians’ characteristics, such as the relationship with children, education level, marital status, employment status, habitual residence, and annual household income; (2) Guardians’ HPV Perceptions, including HPV and HPV vaccine knowledge, HPV susceptibility, and HPV infection severity. HPV and HPV vaccine knowledge included aspects such as HPV transmission routes, cervical cancer, HPV vaccines, etc. Guardians’ HPV knowledge scores were categorized as high- or low-level based on whether their total score exceeded 5 points; (3) Guardians’ Vaccine Attitude, including the importance, effectiveness, and safety of the HPV vaccine, and the comparison between actual age of girls and intended vaccination age of guardians; (4) Cues to Action, including anyone surrounding me diagnosed with cervical cancer or who has received HPV vaccine, whether they had heard negative news about HPV vaccine, and whether doctors recommended HPV vaccine; (5) The channels of obtaining information for HPV and HPV vaccines; (6) Guardians’ HPV vaccine preference and reasons for refusing HPV vaccines. All questions were assessed using a 5-point Likert scale (strongly agree, agree, neutral, disagree, strongly disagree) or dichotomous items (yes/no). The questionnaire is shown in Appendix A.

### 2.4. Statistical Analysis

Statistical analyses were performed using R 4.3.2 software (R Foundation for Statistical Computing, Vienna, Austria). As for categorical variables, frequencies or percentages were presented. The differences between intend and refuse vaccination groups were analyzed using the Chi-square test or the Wilcoxon rank sum test. Univariate analyses were used to explore the factors associated with guardians’ HPV vaccine acceptance for girls. To illustrate the influence of each step of the HBM on the guardians’ HPV vaccine acceptance for girls, the forward stepwise method was applied to conduct the multivariable logistic regression model. The Hosmer and Lemeshow test was used to assess the goodness of fit for each step. Values of *p* < 0.05 were considered statistically significant.

## 3. Results

### 3.1. Demographics and Baseline Characteristics

A total of 2541 guardians were selected to participate in the survey, and 2157 valid questionnaires were retrieved (valid questionnaire rate: 84.89%). Exclusion criteria: a girl’s age was not 9–17 years old (*n* = 64), guardians with the same girl (*n* = 132), guardians with highly repetitive responses (*n* = 188). The demographics and baseline characteristics of participants are shown in Table 1. This study involved 2157 girls with balanced age distribution across three groups (9–11, 12–14, and 15–17 years), each comprising approximately one-third of the total participants. Education levels were predominantly primary school (39.8%), followed by junior high (31.1%) and senior/vocational high school (29.1%). A total of 2059 girls (95.46%) had household registration in Zhejiang province. The most common guardians’ relationship was mother (*n* = 1665, 77.19%). The majority of the guardians’ education level was an undergraduate degree or above (*n* = 1114, 51.65%). The number of guardians without a spouse was relatively low (*n* = 114, 5.29%). Most guardians were full-time employees (*n* = 1590, 73.71%), and most of them were urban residents (*n* = 1574, 72.97%). The largest proportion of the average annual household income was ≥ 100,000 yuan (*n* = 875, 40.57%). Among all participants, 1870 were willing to receive the HPV vaccine for their girls (86.69%).

### 3.2. Guardians’ Awareness of HPV and HPV Vaccine

The guardians’ knowledge score of HPV, cervical cancer, and HPV vaccines is shown in Table 2. Among all participants, 68.1% were aware of the main transmission route of HPV, 91.19% were familiar with cervical cancer, and 87.48% acknowledged that HPV vaccination was the most cost-effective approach for preventing cervical cancer. The correct response rates for the seven questions regarding HPV and vaccine knowledge varied from 27.91% to 91.19%, and the knowledge scores ranged from 0 to 7 points (with a maximum score of 7 points), with an average score of 5.14 points (M = 5.14, SD = 1.69). 1235 guardians (57.26%) fell into the high-knowledge category.

### 3.3. Guardians’ HPV Vaccine Acceptance for Girls Aged 9–17

Univariate analyses on guardians’ HPV vaccine acceptance for girls are shown in Table 3. Univariate analyses demonstrated that guardians’ education degree, employment status, habitual residence, annual household income, level of HPV and HPV-related diseases knowledge, perceived risk of HPV infection in their girls, perceived transmission risk after infection, perceived health consequences of HPV infection, anyone surrounding them diagnosed with cervical cancer or who has received HPV vaccines, regarding HPV vaccination as important for their girl’s health, perceiving HPV vaccines as effectiveness and safety, their girl’s actual age exceeded guardians’ intended vaccination age, having heard less negative information about HPV vaccines, and having received recommendations from doctors have significantly influenced guardians’ acceptance (*p* < 0.05).

Multivariable logistic regression analyses on guardians’ HPV vaccine acceptance for their girls are shown in Table 4. In the final step, guardians with a family income ranging from 50,000 to 99,999 yuan (OR = 1.70, 95%CI = 1.03–2.80), or ≥100,000 yuan (OR = 1.94, 95%CI = 1.14–3.32), those who thought their children had an average or high risk of HPV infection (OR = 1.74, 95%CI = 1.04–2.93; OR = 3.15, 95%CI = 1.40–7.10), those who regarded HPV vaccination as very important for their children’s health (OR = 6.70, 95%CI = 1.61–27.83), those with anyone surrounding them who had received HPV vaccines (OR = 2.03, 95%CI = 1.41–2.92), and those who had received recommendations from doctors (OR = 2.32, 95%CI = 1.65–3.25) were statistically significant increased acceptance of HPV vaccine for girls among guardians. Guardians with a senior high school education (OR = 0.57, 95%CI = 0.37–0.87), those whose girl’s actual age was less than intended vaccination age (OR = 0.49, 95%CI = 0.32–0.75), and those who had heard negative news about HPV vaccines (OR = 0.59, 95%CI = 0.43–0.82) were statistically significantly associated with decreased acceptance of HPV vaccine for girls among guardians.

### 3.4. The Channels of Obtaining Information on HPV and HPV Vaccine

The channels of obtaining information for HPV and HPV vaccines are shown in Figure 2. Findings from the investigation indicated that guardians most frequently preferred obtaining HPV vaccination-related information through government or CDC platforms (*n* = 673, 31.20%), doctor or medical staff (*n* = 288, 13.35%), and social media platforms (*n* = 280, 12.98%) (Figure 2A). Regarding preferences for public education platforms, guardians overwhelmingly favored accessing HPV vaccine-related educational content through government or CDC platforms (*n* = 1494, 69.26%), WeChat platforms (*n* = 743, 34.45%), and medical knowledge platforms (*n* = 701, 32.50%) (Figure 2B).

### 3.5. Guardians’ HPV Vaccine Preference and Reasons for Refusing HPV Vaccine

Guardians’ HPV vaccine preference and reasons for refusing HPV vaccines are shown in Figure 3. In terms of HPV vaccine preference, the majority of guardians expressed willingness to have their girl receive both domestic and imported HPV vaccines (*n* = 868, 46.42% selected “both acceptable”), with a pronounced preference for the 9-valent HPV vaccine (*n* = 979, 52.39%) (Figure 3A). Among guardians reluctant to vaccinate their daughters with HPV vaccines (*n* = 287, 13.31%), the primary reasons cited were concerns about vaccine safety (*n* = 94, 32.75%), the belief that the child was too young for immediate vaccination (*n* = 89, 31.01%), and the high cost of HPV vaccines (*n* = 29, 10.10%) (Figure 3B).

## 4. Discussion

Guardians’ intentions play a pivotal role in determining HPV vaccination among adolescent girls [17]. In this study, guardians exhibited a high acceptance rate (86.7%) of the HPV vaccine for adolescent girls. This rate is significantly higher than the previous study in China (46.3%) [18] and another study in Zhejiang (56.5%) [19]. As an economically advanced province in eastern China, Zhejiang’s growing HPV vaccine acceptance among guardians can be attributed to the widespread dissemination of HPV vaccine-related knowledge [20].

Demographic characteristics were examined as fundamental determinants of HPV vaccination decisions. The multivariable analyses indicated that guardians with higher incomes exhibited significantly stronger HPV vaccine acceptance, consistent with findings from other studies [21]. As the HPV vaccine has not been included in the Chinese NIP, guardians must bear the full cost of HPV vaccination for adolescent girls. Thus, we advocate that the HPV vaccine be included in the NIP or implement financial subsidies and preferential policies to enhance HPV vaccination coverage among adolescent girls, ensuring accessible pricing for economically disadvantaged groups is critical [22].

Guardians’ HPV perceptions were assessed to quantify their disease risk awareness in relation to vaccine acceptance. Although guardians’ HPV knowledge showed no statistically significant difference in HPV vaccine acceptance when other factors were included in multivariable analyses, guardians’ HPV knowledge still revealed a positive association between higher HPV knowledge levels and stronger HPV vaccine acceptance in univariate analyses. This suggests that with widespread HPV and HPV vaccine education, guardians generally had a better understanding, making HPV knowledge no longer an influencing factor [23]. Also, perceiving their girls as being at higher risk of HPV infection significantly increased guardians’ HPV vaccine acceptance for girls, consistent with findings from other studies [24]. Thus, tailored intervention messages on HPV susceptibility for guardians are more effective in enhancing health literacy regarding HPV vaccination among adolescent girls than conventional HPV public education [25].

Guardians’ vaccine attitude was quantified as a critical determinant of guardian-mediated vaccine hesitancy. This step exerted the strongest influence on HPV vaccine acceptance. The multivariable analyses showed that perceiving HPV vaccination as important for their girls’ health significantly increased guardians’ HPV vaccine acceptance for girls. While belief in vaccine efficacy and safety is often cited as a symbol of vaccine confidence [26], due to the high safety profile of licensed HPV vaccines in China, supported by long-term post-market surveillance [27], these factors showed no statistically significant difference when they were included in multivariable analyses. In this context, guardians’ HPV vaccine acceptance appears to be driven by rational health assessments and risk prevention awareness. Additionally, this study found a positive correlation between girls’ age and guardians’ vaccination intention age, with younger girls’ guardians being more reluctant to vaccination, consistent with findings from other studies [28]. This suggests that guardians intend to allow their girls to receive the HPV vaccine until they are older. Therefore, emphasizing the importance of timely vaccination during the optimal age window may be a key strategy to enhance guardians’ HPV vaccine acceptance.

Cues to action were analyzed as accelerators of the vaccine decision-making process. The multivariable analyses indicated that guardians who had not heard about negative news about HPV vaccines or had social contacts who received the HPV vaccines exhibited stronger HPV vaccine acceptance. Another study shows that relatives and friends play a pivotal role in shaping guardians’ attitudes, with vaccinated individuals alleviating others’ distrust [29], which is consistent with our study. Doctor or medical staff recommendations also significantly boosted guardians’ HPV vaccine acceptance for girls, consistent with multiple studies demonstrating the positive impact of medical advice on vaccination decisions [30]. This highlights the critical role of healthcare providers in health education for promoting HPV vaccination and reducing cervical cancer incidence [31]. Therefore, when designing intervention strategies, environmental factors could impact guardians’ HPV vaccine acceptance, with health education by medical personnel serving as a guide. Engaging HPV vaccine recipients to collectively disseminate information can further strengthen vaccine confidence among information recipients.

In addition to interpersonal communication, digital health interventions also have been shown to reduce HPV vaccines hesitancy [32], with sustained media exposure influencing public immunization trends [33]. This study found that guardians most expected to receive HPV or HPV vaccines information from government or CDC platforms, doctor or medical staff, and social media platforms. Additionally, guardians preferred digital channels like government or CDC platforms, WeChat platforms, and medical knowledge platforms to obtain public education on HPV vaccination. This indicates that guardians exhibit a stronger preference for receiving information disseminated by authoritative institutions, consistent with findings from other studies [34]. Additionally, our study revealed no statistically significant differences in channels for obtaining HPV information between guardians’ HPV vaccine acceptance for girls. This may be attributed to the mixed positive and negative influences of HPV vaccine information from different channels [35]. This finding emphasizes the need for future studies to dissect how different information channels either facilitate or hinder HPV vaccine acceptance, rather than limiting the focus on access to information.

In terms of guardians’ HPV vaccine preferences, over 50% guardians preferred the 9-valent HPV vaccine for their daughters. This suggests that guardians preferred higher-valent HPV vaccines for their girls, consistent with multiple studies [18,24]. Among guardians’ reasons for refusing the HPV vaccine, the primary reasons cited were concerns about vaccine safety, the belief that their girls were too young, and the high cost of the HPV vaccine, consistent with findings from other studies [36,37]. To advance the WHO’s “90-70-90” targets, tailored health education interventions are essential to build guardians’ confidence in HPV vaccines, increase girls’ vaccination rates, and ultimately reduce cervical cancer incidence.

### Limitation

This study has several limitations. First, this study was a cross-sectional design, only allowing for correlation analysis between variables; causal inference cannot be established from the current data. Second, participants may provide answers they view as ideal, which may inflate self-reported vaccination intent due to social desirability bias. Furthermore, this study was commenced in May 2024, prior to China’s immunization program strategy required that HPV vaccines should be prioritized for female recipients. The sample comprised guardians of adolescent girls only, limiting generalizability for guardians of adolescent boys. Additionally, while HPV vaccination decisions are multifactorial and guardians’ attitudes may be influenced by adolescent girls’ own preferences, this study did not fully account for adolescent girls’ vaccination acceptance. Finally, the study sample consisted of individuals who voluntarily enrolled through vaccination clinics, which inherently represents a population with more positive attitudes toward HPV vaccines and higher vaccination intent, introducing potential volunteer bias.

## 5. Conclusions

This study demonstrates that guardians of adolescent girls in Zhejiang Province had a good level of awareness regarding HPV and HPV vaccines. HPV vaccine acceptance was 86.7%, which was associated with guardians’ household income, perceived HPV infection risk in girls, perceived health importance of the vaccine, vaccination status of friends, awareness of negative vaccine information, and medical staff recommendations. Remarkably, we found Chinese guardians preferred to vaccinate their daughters at the age of 17–18 years, which was later than the WHO’s recommended optimal age period (9–14 years), coupled with their safety concerns, contributing to vaccine hesitancy. In addition, guardians preferred to obtain digital information on the vaccine via government or CDC platforms, WeChat platforms, and medical knowledge platforms. At the same time, we have established an intervention cohort by delivering health messages monthly on vaccines and related diseases through government or CDC official channels for 6 months. Future evaluations will be conducted to explore the effects of digital intervention on vaccination compliance using a follow-up questionnaire from guardians. This will provide a more robust theoretical foundation for implementing digital health interventions in vaccination promotion.

## Figures and Tables

**Figure 1 vaccines-13-00840-f001:**
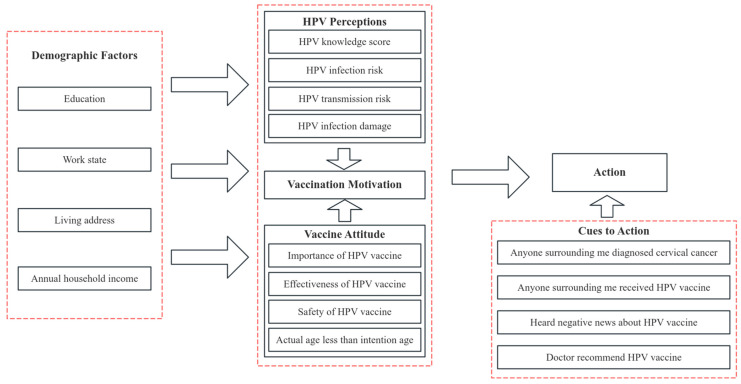
Health Belief Model of HPV vaccine acceptance for 9–17 girls among guardians.

**Figure 2 vaccines-13-00840-f002:**
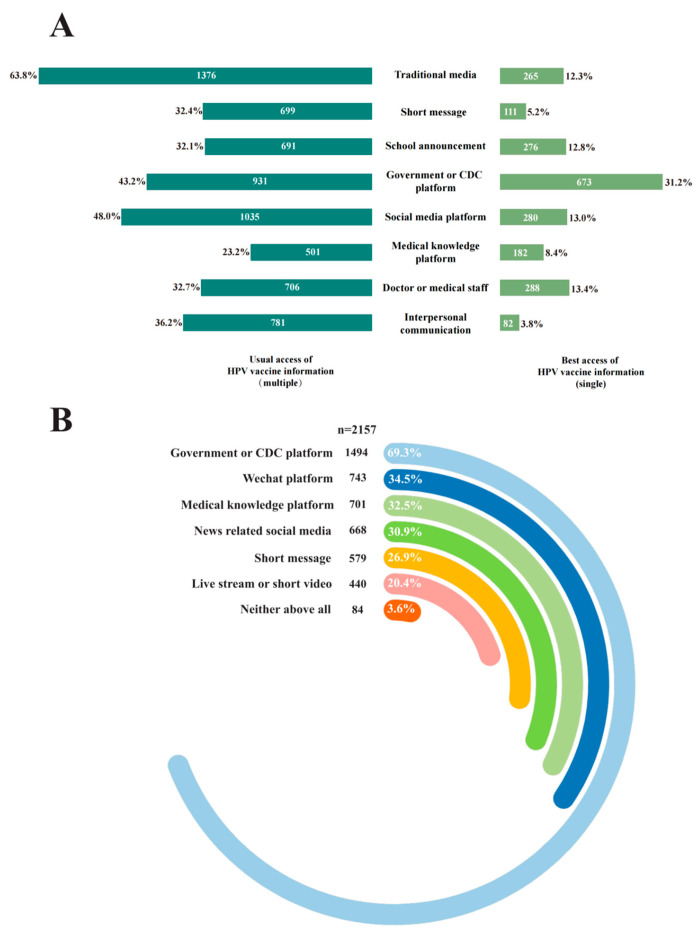
The channels of obtaining information for HPV and HPV vaccines. (**A**) The most preferred channel to receive vaccine-related information; (**B**) The most preferred public platform to receive HPV-related educational information.

**Figure 3 vaccines-13-00840-f003:**
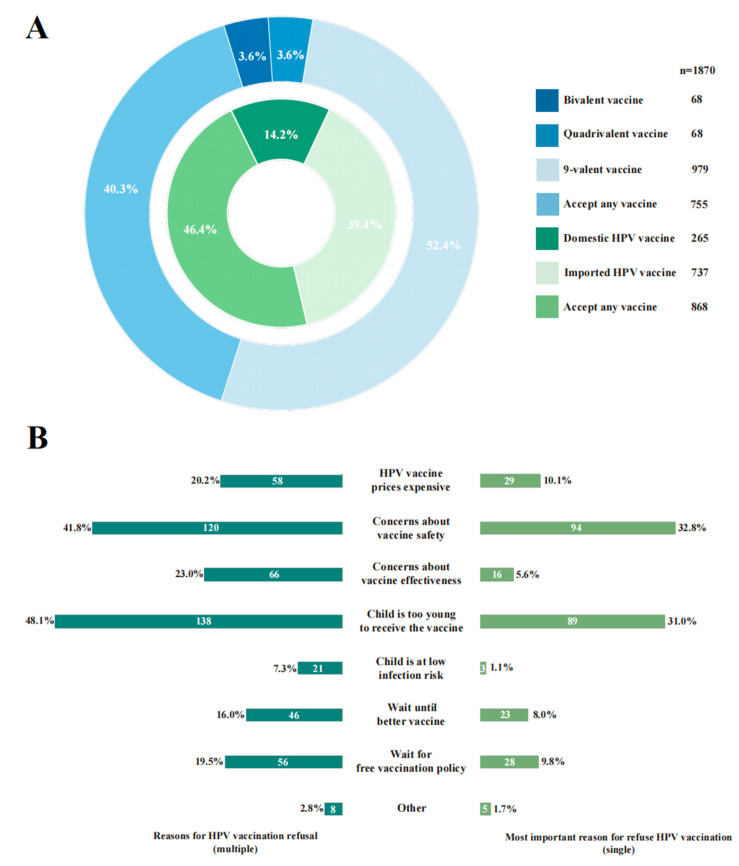
Guardians’ HPV vaccine preference and reasons for vaccine hesitancy. (**A**) Guardians’ HPV vaccine preference. (**B**) Reasons for HPV vaccine hesitancy among guardians.

**Table 1 vaccines-13-00840-t001:** Demographics and baseline characteristics of participants.

Characteristics	Cases (*n*)	Proportion (%)
**Children’s characteristics**		
Age		
9 to 11 years old	722	33.47
12 to 14 years old	710	32.92
15 to 17 years old	725	33.61
Children’s education		
Primary school	859	39.82
Junior high school	670	31.06
Senior or vocational high school	628	29.11
Household registration province		
Zhejiang province	2059	95.46
Other province	98	4.54
**Guardians’ characteristics**		
Relationship with children		
Father	466	21.60
Mother	1659	76.91
Grandparents	3	0.14
Other	29	1.34
Education		
Junior high school or lower	562	26.05
Senior high school	481	22.30
Undergraduate or above	1114	51.65
Marital status		
Married	2043	94.71
Unmarried	25	1.16
Other (including divorced or widowed)	89	4.13
Employment status		
Full-time job	1665	77.19
Part-time job	147	6.82
Housework and unemployment	302	14.00
Other	43	1.99
Habitual residence		
Urban	1574	72.97
Rural	583	27.03
Annual household income, CNY (USD)		
Less than 20,000 (2789)	221	10.25
20,000–49,999 (2789–6972)	410	19.01
50,000–99,999 (6972–13,944)	651	30.18
More than 100,000 (13,944)	875	40.57
HPV vaccine acceptance	1870	86.69

**Table 2 vaccines-13-00840-t002:** Guardians’ knowledge of HPV, cervical cancer, and HPV vaccines.

Variables	Correct Cases (*n*)	Proportion (%)
Know HPV transmission routes	1469	68.10
Know cervical cancer	1967	91.19
Know HPV vaccine could defend against cervical cancer most effectively	1887	87.48
Know recommended population for HPV vaccination	602	27.91
Know HPV vaccine valence	1812	84.01
Know optimal HPV vaccination time	1429	66.25
Know cervical cancer screening	1928	89.38
Average score (Mean ± SD)	5.14 ± 1.69	

**Table 3 vaccines-13-00840-t003:** Univariate analyses on HPV vaccine acceptance for girls aged 9–17 among guardians.

Variables	Total (*n* = 2157)	Refuse to Vaccinate (*n* = 287)	Intend to Vaccinate (*n* = 1870)	Statistic	*p*
**HPV knowledge**					
HPV knowledge score				χ^2^ = 65.85	<0.001
Low (0–5 points)	922 (42.74)	186 (20.17)	736 (79.83)		
High (6–7 points)	1235 (57.26)	101 (8.18)	1134 (91.82)		
**Questions related to disease**					
HPV infection risk				U = 218,543.00	<0.001
Very low	640 (29.67)	117 (18.28)	523 (81.72)		
Low	415 (19.24)	53 (12.77)	362 (87.23)		
Neutral	716 (33.19)	95 (13.27)	621 (86.73)		
High	239 (11.08)	14 (5.86)	225 (94.14)		
Very high	147 (6.82)	8 (5.44)	139 (94.56)		
HPV transmission risk				U = 229,555.50	<0.001
Very low	687 (31.85)	120 (17.47)	567 (82.53)		
Low	366 (16.97)	38 (10.38)	328 (89.62)		
Neutral	597 (27.68)	88 (14.74)	509 (85.26)		
High	284 (13.17)	25 (8.80)	259 (91.20)		
Very high	223 (10.34)	16 (7.17)	207 (92.83)		
HPV infection damage				U = 213,103.00	<0.001
Absolutely not severe	55 (2.55)	13 (23.64)	42 (76.36)		
Not severe	36 (1.67)	11 (30.56)	25 (69.44)		
Neutral	332 (15.39)	73 (21.99)	259 (78.01)		
Severe	681 (31.57)	87 (12.78)	594 (87.22)		
Very severe	1053 (48.82)	103 (9.78)	950 (90.22)		
Anyone surrounding me diagnosed with cervical cancer				χ^2^ = 7.22	0.007
No	1621 (75.15)	234 (14.44)	1387 (85.56)		
Yes	536 (24.85)	53 (9.89)	483 (90.11)		
**Questions related to vaccine**					
Anyone surrounding me received HPV vaccine				χ^2^ = 130.48	<0.001
Nobody	379 (17.57)	119 (31.40)	260 (68.60)		
Myself or someone close to me	1778 (82.43)	168 (9.45)	1610 (90.55)		
Importance of HPV vaccine				U = 124,015.50	< 0.001
Very unimportant	15 (0.70)	8 (53.33)	7 (46.67)		
Unimportant	34 (1.58)	25 (73.53)	9 (26.47)		
Neutral	195 (9.04)	94 (48.21)	101 (51.79)		
Important	605 (28.05)	94 (15.54)	511 (84.46)		
Very important	1308 (60.64)	66 (5.05)	1242 (94.95)		
Effectiveness of HPV vaccine				U = 141,993.50	<0.001
Very ineffective	5 (0.23)	4 (80.00)	1 (20.00)		
Ineffective	18 (0.83)	11 (61.11)	7 (38.89)		
Neutral	606 (28.09)	172 (28.38)	434(71.62)		
Effective	853 (39.55)	72 (8.44)	781 (91.56)		
Very effective	675 (31.29)	28 (4.15)	647 (95.85)		
Safety of HPV vaccine				U = 137,732.50	<0.001
Very unsafe	4 (0.19)	3 (75.00)	1 (25.00)		
Unsafe	20 (0.93)	11 (55.00)	9 (45.00)		
Neutral	586 (27.17)	178 (30.38)	408 (69.62)		
Safe	932 (43.21)	69 (7.40)	863 (92.60)		
Very safe	615 (28.51)	26 (4.23)	589 (95.77)		
Actual age less than intended age					
Yes	1560 (72.32)	256 (16.41)	1304 (83.59)	χ^2^ = 47.10	<0.001
No	597 (27.68)	31 (5.19)	566 (94.81)		
Heard negative news about HPV vaccine				χ^2^ = 15.13	<0.001
No	1463 (67.83)	166 (11.35)	1297 (88.65)		
Yes	694 (32.17)	121 (17.44)	573 (82.56)		
Doctor recommended HPV vaccine				χ^2^ = 75.26	<0.001
No	1117 (51.78)	217 (19.43)	900 (80.57)		
Yes	1040 (48.22)	70 (6.73)	970 (93.27)		

χ^2^: Chi-square test, U: Wilcoxon rank sum test.

**Table 4 vaccines-13-00840-t004:** Multivariable logistic regression analyses on HPV vaccine acceptance for girls aged 9–17 among guardians.

Variables	Step 1		Step 2		Step 3		Step 4	
	*p* Value	OR [95%CI]	*p* Value	OR [95%CI]	*p* Value	OR [95%CI]	*p* Value	OR [95%CI]
**Demographic Variables**								
Education								
Junior high school or lower		1.00 (Reference)		1.00 (Reference)		1.00 (Reference)		1.00 (Reference)
Senior high school	0.068	0.73 (0.52–1.02)	0.014	0.64 (0.45–0.91)	0.024	0.63 (0.42–0.94)	0.009	0.57 (0.37–0.87)
Undergraduate or above	0.248	1.25 (0.86–1.81)	0.985	1.00 (0.67–1.47)	0.822	0.95 (0.61–1.48)	0.348	0.80 (0.50–1.27)
Employment status								
Full-time job		1.00 (Reference)		1.00 (Reference)		1.00 (Reference)		1.00 (Reference)
Part-time job	0.947	1.02 (0.62–1.65)	0.893	1.04 (0.63–1.71)	0.222	1.43 (0.81–2.52)	0.246	1.41 (0.79–2.53)
Housework and unemployment	0.128	0.76 (0.53–1.08)	0.270	0.81 (0.56–1.18)	0.999	1.00 (0.65–1.53)	0.935	1.02 (0.66–1.57)
Other	0.267	1.82 (0.63–5.20)	0.197	2.05 (0.69–6.12)	0.433	1.63 (0.48–5.51)	0.342	1.85 (0.52–6.59)
Habitual residence								
Rural		1.00 (Reference)		1.00 (Reference)		1.00 (Reference)		1.00 (Reference)
Urban	0.800	0.96 (0.71–1.30)	0.455	0.89 (0.65–1.21)	0.301	0.83 (0.58–1.18)	0.232	0.80 (0.55–1.15)
Annual household income (CNY)								
Less than 20,000		1.00 (Reference)		1.00 (Reference)		1.00 (Reference)		1.00 (Reference)
20,000–49,999	0.068	1.46 (0.97–2.18)	0.156	1.36 (0.89–2.06)	0.141	1.43 (0.89–2.32)	0.204	1.38 (0.84–2.26)
50,000–99,999	<0.001	2.18 (1.45–3.29)	0.002	1.94 (1.27–2.98)	0.012	1.86 (1.14–3.02)	0.036	1.70 (1.03–2.80)
More than 100,000	<0.001	2.39 (1.54–3.72)	<0.001	2.19 (1.39–3.45)	0.010	1.99 (1.18–3.34)	0.015	1.94 (1.14–3.32)
**HPV Perceptions**								
HPV knowledge score								
Low				1.00 (Reference)		1.00 (Reference)		1.00 (Reference)
High			<0.001	2.17 (1.63–2.89)	0.146	1.28 (0.92–1.77)	0.315	1.19 (0.85–1.67)
HPV infection risk								
Very low				1.00 (Reference)		1.00 (Reference)		1.00 (Reference)
Low			0.106	1.45 (0.92–2.29)	0.054	1.66 (0.99–2.77)	0.072	1.61 (0.96–2.70)
Neutral			0.005	1.90 (1.21–2.96)	0.009	1.95 (1.18–3.23)	0.036	1.74 (1.04–2.93)
High			<0.001	3.95 (1.91–8.20)	0.002	3.55 (1.60–7.86)	0.006	3.15 (1.40–7.10)
Very high			0.014	3.15 (1.26–7.87)	0.100	2.39 (0.85–6.75)	0.135	2.25 (0.78–6.55)
HPV transmission risk								
Very low				1.00 (Reference)		1.00 (Reference)		1.00 (Reference)
Low			0.259	1.33 (0.81–2.19)	0.514	1.20 (0.69–2.09)	0.507	1.21 (0.69–2.11)
Neutral			0.711	0.92 (0.58–1.44)	0.510	0.84 (0.50–1.41)	0.732	0.91 (0.54–1.55)
High			0.619	0.86 (0.47–1.56)	0.056	0.52 (0.27–1.02)	0.068	0.53 (0.27–1.05)
Very high			0.682	1.16 (0.58–2.32)	0.794	0.90 (0.41–1.97)	0.854	0.93 (0.42–2.05)
HPV infection damage								
Absolutely not severe				1.00 (Reference)		1.00 (Reference)		1.00 (Reference)
Not severe			0.138	0.47 (0.17–1.28)	0.311	0.53 (0.15–1.82)	0.139	0.38 (0.11–1.36)
Neutral			0.441	0.75 (0.36–1.57)	0.602	0.78 (0.30–2.00)	0.407	0.66 (0.24–1.77)
Severe			0.739	1.13 (0.55–2.31)	0.509	0.73 (0.29–1.83)	0.381	0.65 (0.25–1.70)
Very severe			0.227	1.54 (0.76–3.12)	0.441	0.70 (0.28–1.74)	0.293	0.60 (0.23–1.56)
**Vaccine Attitudes**								
Importance of HPV vaccine								
Very unimportant						1.00 (Reference)		1.00 (Reference)
Unimportant					0.266	0.42 (0.09–1.93)	0.476	0.56 (0.11–2.76)
Neutral					0.724	0.79 (0.21–2.98)	0.818	0.85 (0.21–3.45)
Important					0.259	2.16 (0.57–8.18)	0.233	2.35 (0.58–9.62)
Very important					0.009	6.00 (1.55–23.14)	0.009	6.70 (1.61–27.83)
Effectiveness of HPV vaccine								
Very ineffective						1.00 (Reference)		1.00 (Reference)
Ineffective					0.139	8.93 (0.49–162.46)	0.258	5.56 (0.28–108.59)
Neutral					0.048	13.97 (1.03–189.88)	0.101	9.47 (0.65–138.67)
Effective					0.039	15.58 (1.14–212.07)	0.088	10.32 (0.71–151.00)
Very effective					0.039	15.73 (1.14–216.66)	0.094	10.01 (0.68–148.01)
Safety of HPV vaccine								
Very unsafe						1.00 (Reference)		1.00 (Reference)
Unsafe					0.853	0.76 (0.04–13.31)	0.939	0.89 (0.05–16.98)
Neutral					0.957	0.93 (0.07–12.37)	0.979	1.04 (0.07–15.12)
Safe					0.553	2.21 (0.16–30.18)	0.629	1.95 (0.13–29.27)
Very safe					0.501	2.50 (0.17–35.90)	0.626	1.98 (0.13–31.29)
Actual age less than intended age								
No						1.00 (Reference)		1.00 (Reference)
Yes					<0.001	0.48 (0.32–0.74)	<0.001	0.49 (0.32–0.75)
**Cues to Action**								
Anyone surrounding me diagnosed with cervical cancer								
No								1.00 (Reference)
Yes							0.438	0.86 (0.59–1.26)
Anyone surrounding me received HPV vaccine								
No								1.00 (Reference)
Yes							<0.001	2.03 (1.41–2.92)
Heard negative news about HPV vaccine								
No								1.00 (Reference)
Yes							0.001	0.59 (0.43–0.82)
Doctor recommended HPV vaccine								
No								1.00 (Reference)
Yes							<0.001	2.32 (1.65–3.25)
**Hosmer and Lemeshow test**	χ^2^ (df) = 2.84 (7), *p* = 0.90		χ^2^ (df) = 2.98 (8), *p* = 0.94		χ^2^ (df) = 14.97 (8), *p* = 0.06		χ^2^ (df) = 5.29 (8), *p* = 0.73	

Adjusted multivariable logistic regression using the stepwise method. HBM dimensions were entered as follows: Step 1: Demographic Variables; Step 2: HPV Perceptions; Step 3: Vaccine Attitudes; Step 4: Cues to Action.

## Data Availability

The data presented in this study are available on request from the corresponding author. The data are not publicly available due to ethical restrictions.

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
