# Peer review of "Awareness, Preference, and Acceptance of HPV Vaccine and Related Influencing Factors Among Guardians of Adolescent Girls in China: A Health Belief Model-Based Cross-Sectional Study"

_vaccines, 2025, doi:10.3390/vaccines13080840_

Round 1
Reviewer 1 Report
Comments and Suggestions for Authors
This is a well-written paper investigating HPV vaccine acceptance among parents of adolescent girls in China. A questionnaire survey was developed using the Health Belief Model.
This reviewer suggests only minor revisions:
2.2. Procedures (lines 111-117) - could benefit from additional details on the expert evaluation - CDC experts? how were they identified and contacted? How was the standardized training conducted? Any details from the validation process?
There are a few grammatical errors:
"As the HPV vaccine haven't covered by China's NIP, guardians should afford the full cost for adolescent girl’s HPV vaccination"
As the HPV vaccine has not been included in the Chinese National Immunization Program, guardians must bear the full cost of HPV vaccination for adolescent girls.
Please revise the manuscript for additional grammar and punctuation.
Discussion
There is considerable repetition of results. Please expand the discussion for "no statistically significant differences in HPV information obtaining channels between guardian's HPV vaccine acceptance for girls" (lines 294-295). The authors discuss "This may be attributed to the mixed positive and negative influences of HPV vaccine information from different channels," and the reference is from a study on the influenza vaccine. if possible, please add a reference and a study related to HPV vaccination. If not, please discuss how this study and reference #35.
Reviewer 2 Report
Comments and Suggestions for Authors
This study Zheng, et al explore the knowledge and attitude of guardians of eligible daughters toward administering HPV vaccine. The authors dud a good job including a large number of participants. However, several major and minor issues need to be addressed.
1. Line 27, 148, 192, 205, and 240: Replace "multivariate" with "multivariable" as the latter is the correct term. See this article to understand the difference between the two statistical tests: https://pmc.ncbi.nlm.nih.gov/articles/PMC3518362/
2. Line 28: Edit "Guardian" to "The guardians"
3. Replace all "~" by a regular dash "-" in the abstract, in section 3.3 in the main text and in all the tables.
4. Line 97: How did you determine that the vaccination rates were comparable? Did you use data from the national registry of each region? Please clarify.
5. Why did you not also assess parents of boys since WHO recommends HPV vaccine for boys as well as girls? Please clarify in section 2.1.
6. Line 98: I find it a bit biased to recruit participants from vaccination clinics since this will include people who are already willing to receive vaccines. As such, this would exclude people who are against receiving vaccines or hesitant to receive them. Additionally, did the authors assess whether any of these participants were already visiting the vaccination clinic to receive the HPV vaccine for themselves or their daughters? Again, this may impact the results of your study and perhaps explains the high vaccine acceptance rate. Please discuss the potential of bias in the limitations paragraph.
7. Section 2.2: did you also run a pilot phase to do a test retest reliability assessment and to ensure that the questions are understandable by the public?
8. Section 2.3.2: Please cite Appendix A here.
9. Line 144: Please provide the name of the developer of the software and the country.
10. Section 2.4: Please indicate the P value used to determine statistical significance.
11. Line 154: What makes the remaining questionnaires invalid? Please clarify.
12. Line 180: Replace "Single factor" with "Univariate." Also, this analysis was not described in section 2.4. Please add it.
13. Table 3: For the "Statistic" column, it's better to replace it with P values as it's more understandable.
14. You didn't explain in the methods (or at least outline in Table 3) what makes a score very low, low, neutral, high, or very high. Either describe It in the methods section or put it in between parentheses next to each score in Table 3.
15. Line 205: Explain in the table title that these are the impact of each factor on HBM step. This is because the column titles "Step 1" , "Step 2", and "Step 3" are unclear and the table should stand alone without the need to refer to the main text to understand what these steps mean.
16. Section 3.5: Here, did you assess the knowledge of the participants regarding each type of the HPV vaccine? Typically, the public are unaware of the different types of the HPV vaccines regarding availability and coverage.
17. Figure 3B: Correct "Child is too young to vaccine" to "Child It's too young to receive the vaccine." Also correct "Reasons for refuse HPV vaccination" to "Reasons for HPV vaccination refusal."
18. I noticed in Figure 3B the fact that the vaccine is expensive is one of the factors that makes the guardians hesitant to administer it to their daughters. Does that mean that the vaccine is not offered by the Chinese government for free? Please explain this in the introduction so that readers are aware beforehand that people need to pay for the vaccine and that this factor might impact their decision regarding vaccination. Also, it would be helpful to include the approximate price of the vaccine in US dollars so that the reader can correlate such information with the household income listed in Table 1.
19. Discussion: Almost every paragraph of the discussion section starts with "In the aspect of" please revise the language and edit these parts to avoid redundancy.
20. Section 4.1: One important very common limitation of cross-sectional survey-based studies is social desirability bias, where participants may provide answers that they perceive as ideal. Read more about it here: https://www.sciencedirect.com/topics/psychology/social-desirability-bias
Reviewer 3 Report
Comments and Suggestions for Authors
Dear Authors,
The aim of this article is not just essential, but also has the potential to make a significant impact in the field.
You did colossal work. The manuscript could be accepted after minor revisions are made.
155-158 lines. The entire text (155-158 lines) reproduces the data in the table exactly. I recommend submitting a summary of the most critical data.
163 line. There is a data discrepancy in row 163, with n = 1,547, and in Table 1, with n = 1,574. The discrepancy is in the total count of the sample. Please specify which number is correct.
Table 3. You provided the coefficients of the statistical method used, but you did not give the p-value or indicate which data are different. The p-value is crucial as it indicates the significance of the results. Please provide this information.
Table 4. This table contains many results and is therefore difficult to "read/understand". It may be appropriate to present only the results of Step 4.
Round 2
Reviewer 2 Report
Comments and Suggestions for Authors
Excellent revision. Thanks to the authors for revising their manuscript and addressing the comments appropriately. I endorse the manuscript for publication.